# Effects of Accelerated Weathering on Degradation Behavior of Basalt Fiber Reinforced Polymer Nanocomposites

**DOI:** 10.3390/polym12112621

**Published:** 2020-11-06

**Authors:** Ummu Raihanah Hashim, Aidah Jumahat, Mohammad Jawaid, Rudi Dungani, Salman Alamery

**Affiliations:** 1Faculty of Mechanical Engineering, Universiti Teknologi MARA (UiTM), Shah Alam 40450, Selangor, Malaysia; ummuraihanahhashim@gmail.com; 2Institute for Infrastructure Engineering Sustainable and Management (IIESM), Universiti Teknologi MARA, Shah Alam 40450, Selangor, Malaysia; 3Department of Biocomposite Technology, Institute of Tropical Forestry and Forest Products, Universiti Putra Malaysia, UPM Serdang 43400, Selangor, Malaysia; 4School of Life Science and Technology, Institut Teknologi Bandung, Bandung 40132, Indonesia; rudi@sith.itb.ac.id; 5Department of Biochemistry, College of Science, King Saud University, P.O. Box 22452, Riyadh 11451, Saudi Arabia; salamery@ksu.edu.sa

**Keywords:** UV exposure degradation, accelerated weathering, polymer, nanocomposites, basalt fiber reinforced polymer composites

## Abstract

This work aims to give insight on the effect of accelerated weathering, i.e., the combination of ultraviolet (UV) exposure and water spraying, on the visual and mechanical properties of basalt fiber reinforced polymer (BFRP) composites. The solvent exchange method, sonication and high shear milling technique were used to prepare the nanocomposite laminates. Three types of laminates were fabricated, i.e., unmodified BFRP, nanosilica modified BFRP and graphene nanoplatelet (GNP) modified BFRP composites with the total fiber loading of 45 wt.%. Glass fiber reinforced polymer (GFRP) laminate was also prepared for performance comparison purposes between the natural and synthetic fibers. The laminates were exposed to UV with a total weathering condition of 504 h using a Quantum-UV accelerated weathering tester. The weathering condition cycle was set at 8 h 60 °C UV exposure and 4 h 50 °C condensation. The discoloration visual inspection on the tested specimen was observed under the optical microscope. The obtained results showed that the UV exposure and water absorption caused severe discoloration of the laminates due to photo-oxidation reaction. The effect of weathering conditions on tensile and flexural properties of unmodified BFRP composites indicated that the UV exposure and water absorption caused reduction by 12% in tensile strength and by 7% in flexural strength. It is also found that the reduction in tensile and flexural properties of nanomodified BFRP composites was smaller than the unmodified system. It concluded from this work, that the mineral based composites (i.e., BFRP) has high potential for structural applications owing to its better properties than synthetic based composites (i.e., GFRP).

## 1. Introduction

Synthetic carbon and glass fiber reinforced polymer composites have been broadly used in marine and civil structures with concrete as structural piers and pile structures because of their high strength-to-weight ratio, flexible design and good corrosion resistance. These fiber-reinforced polymer (FRP) composites are effective materials for concrete as it enables them to be an option for steel reinforcement for engineering applications [1,2]. However, one of the most influential factors which limits their application is their high initial cost as well as the high energy consumption of production. Global synthetic fiber prices increased by almost 13% in 2017 which is triple 2016’s rate of increase and it is the biggest annual rise since 2009 [3]. The use of natural fiber as an alternative to synthetic fiber has been found to be an effective way to solve related issues. Other than cost, increasing public concern about the environment, climate change, energy consumption and the greenhouse effect, as well as demand for environmentally friendly materials has captured the attention of industry and academia to use natural fibers as a substitution for glass and carbon fiber. [4,5,6,7]. To date, the global natural fiber composites market is estimated to touch USD 10.89 billion due to the increasing demand for lightweight products from the automotive industry and the increasing consciousness of green products and environmental sustainability [8]. 

Natural fibers are biobased materials originating from renewable resources. They are nonabrasive biodegradable materials, with less energy-consumption and a low health risk, which often have low densities and lower processing costs than synthetic fibers [9,10,11,12,13]. They are divided into two different categories; organic and inorganic natural fibers. Organic natural fibers are then further classified into plant-based and animal-based natural fiber. The use of plant-based fibers includes those from the (i) seed: cotton, (ii) stems: kenaf, hemp, jute, flax, (iii) fruit hair: kapok, (iv) leaves: pineapple, banana, sisal, and others. Animal-based natural fibers are commonly used in textile industries as clothing materials (silk) as well as on a daily basis (horsehair) for brushes and other items [14,15]. Meanwhile, inorganic natural fibers refer to mineral-based fibers such as asbestos and basalt fibers [13,16,17,18,19,20]. However, asbestos is now banned in many countries due to the hazard cause by this material which could lead to a few health problems such as cancer and asbestosis [21]. Basalt fibers are made of volcanic rocks which have been through the single melt process at 1500 °C, followed by extrusion of the molten rock into filaments. The extrusion process of basalt fibers is a more energy saving process and simpler than any other relative fibers [22]. 

Basalt fiber has shown great potential as an alternative material to glass fiber due to its lower cost and good mechanical properties. This fiber also offers good thermal stability, good sound and thermal insulation properties, superior abrasion resistance and is more durable compared to glass fiber especially at high temperature [23,24]. With a density of between 2.7 to 2.8 g/cm^3^, basalt fibers have similar properties to glass fiber with the modulus of elasticity range between 89 and 95 GPa, tensile strength between 2800 and 4900 MPa, and the ultimate tensile strain between 3.00 and 5.00% [25]. Basalt fibers are a nonreactive reinforcement material which is also stable, biodegradable, abundantly available, inert and eco-friendly [26,27]. This mineral-based fiber is taken from rocks which owe the reason for their different chemical contents to their geographical distribution. It is chemically rich with oxides, magnesium, calcium, sodium, potassium, silica, iron and alumina [28]. These fibers give an alternative option to commercial materials of equivalent performance. 

Despite the advantages of this BFRP composite, the use of biocomposites for external applications is not prevalent due to the degradation risk when exposed to different environmental conditions. The durability of these materials, especially in continuous hot and humid environments is one of the most important issues that limit its versatility [11]. The biocomposites are prone to suffer from the deterioration of several properties after weathering periods, such as discoloration, loss of mechanical and thermal stability, and surface roughness [29]. There are many weathering factors that affect the performance of FRP composites. Direct sunlight exposure will lead to the photo-degradation process, which eventually can modify the surface chemistry of the composites [15]. The photo-degradation mechanism is stimulated by ultraviolet (UV) irradiation and oxygen concentration on the surface of the composite as the UV exposure penetrates a few microns of the material’s surface [15,30,31]. This results in the breakage of polymer chains producing free radicals and reducing the molecular weight of the polymers. This would cause the surface degradation and reduce the mechanical properties of the material, where flaws resulting from degradation can serve as stress concentrators [31]. 

The degradation behavior and durability studies of natural FRP composites have been investigated by several researchers. Dayo et al. [32] investigated the effect on the mechanical properties of silane treated hemp fiber (STHF) composites after exposure to accelerated weathering conditions at selected relative humidity (RH) until saturated moisture uptake. Chee et al. [29] considered the influence of environmental effects through accelerated weathering conditions on the degradation behavior and thermal stability of hybrid bamboo/kenaf fiber-reinforced polymer composites. In another study, Butylina [33] studied the effects of outdoor exposure on the deterioration properties of wood-polypropylene composites especially on the aspects of color stability and impact strength. Most of the studies reported degradation behavior of the organic natural fiber-reinforced polymer composites after UV exposure. To date, investigation of the durability of inorganic fibers, especially on basalt fiber reinforced polymer (BFRP) composites subject to UV exposure and water spraying weathering conditions are still limited. Therefore, in this work, the effect of accelerated weathering conditions, which consisted of UV irradiation and water spraying, on the mechanical and physical properties of BFRP composite was investigated. The assessments on the stress-strain response of BFRP composite were then evaluated to compare with the commercial glass fiber reinforced polymer (GFRP). The tensile and flexural test was conducted to determine the mechanical properties of the composites.

## 2. Materials and Methods 

### 2.1. Materials

Commercial epoxy resin Miracast 1517 (Part A and B) was supplied by Miracon (M) Sdn Bhd, Seri Kembangan, Malaysia. This epoxy resin is a low viscosity epoxy laminating resin suitable to be used for composite laminates which can meet all the high-performance properties of composite structures. The unidirectional basalt fiber and glass fiber fabrics with a density of 2.6 g/cm^3^ and 2.7 g/cm^3^, respectively, were supplied by Innovative Pultrusion Sdn. Bhd, Seremban, Malaysia. The graphene nanoplatelet (GNP) used in this study was XGnP-Graphene Nanoplatelets Grade M5 in the form of black granules manufactured by XG Sciences, Lansing, MI, USA with an average thickness of 6−8 nm, average particle diameter of 5 µm, and a typical surface area of 120 to 150 m^2^/g. Nanopox F400 nanosilica gel manufactured by Evonik Industries AG, Essen, Germany was used to be embedded in epoxy resin. It is a high performance, versatile, silica-reinforced bisphenol-A based epoxy resin for use in fiber composites. It consists of modified SiO_2_ with a very small particle size of 20 nm (which is good to penetrate the tightly meshed fibers) and narrow particle-size distribution (maximum diameter of 50 nm). Despite the high silica content of 40 wt.%, this nanosilica gel, comparatively, has low viscosity due to agglomerate-free colloidal dispersion of the nanoparticles in the resin.

### 2.2. Fabrication of Composites

Initially, unmodified and nanomodified epoxy resin was prepared using the Miracast 1517 epoxy resin. For the unmodified epoxy to be used reinforced with fiber, the epoxy (Part A) was weighed according to the amount required and then mixed with hardener (Part B). For the nanomodified epoxy resin, ethanol was used as the solvent to disperse the GNP, meanwhile, nanosilica was dispersed using the mechanical stirrer. The BF and GF fabrics were cut according to the required dimensions. The unmodified and nanomodified epoxies were weighted according to the required final specimen’s size. The hardener was then added into the epoxy resin at room temperature. Hand lay-up and vacuum bagging method were adopted in the fabrication process, starting with a layer of resin, alternating with a layer of fiber fabric and ending with a layer of resin again until the required thickness had been achieved. Once the thickness was achieved, a perforated release film was put on top of the laminate to allow the excessive resin to pass through followed by absorption by the absorption fabric. The vacuum bagging film was then placed on top of the laminate and pressed firmly against the sealant tape to provide a tight vacuum system with the proper spread as this would result in the formation of wrinkles that would affect the surface finish of the laminate. Finally, the laminate was vacuumed for an hour to remove any air trapped, bubbles and excessive resin before left to cure at room temperature for 24 h. A total of five samples per group of FRP composites were fabricated for testing purposes. Figure 1 shows the example of fabricated unmodified BFRP composite samples. Table 1 lists the details and labels used to represent the samples.

### 2.3. Accelerated Weathering Test

A QUV accelerated weathering tester by Q-Lab, Westlake, OH, USA was used to perform the weathering test according to ASTM G154. The test was intended to reproduce the weathering effects that occur when materials are exposed to sunlight (either direct or through window glass) and moisture as rain or dew in actual usage. The radiation spectrum centered in the ultraviolet wavelengths was provided using the fluorescent UV lamps. The samples were illuminated by UVA-351 fluorescent lamps with a wavelength of 351 nm and the irradiance level of 0.89 W/m^2^. The temperature in the chamber was controlled by the heaters and the moisture was provided by forced condensation. The fabricated samples of GFRP and BFRP for tensile and flexural tests were mounted in the sample holder in the chamber. The samples were exposed for 504 h consisting of 8 h UV exposure at 60 °C and 4 h condensation at 50 °C. The hot vapors maintained the chamber environment at 100% relative humidity at an elevated temperature. Figure 2 shows the sample setup in the accelerated weathering chamber.

### 2.4. Moisture Absorption and Swelling Test

This test was conducted in accordance with ASTM D570-98 to determine the moisture absorption of the BFRP composites. At first, the unconditioned and conditioned specimens were dried in an oven for a specified period and temperature, and then placed in a desiccator to cool. The specimens were immersed in water for 24 h at room temperature, then were removed, patted dry, and weighed to determine the final weight. For the swelling test, the thickness of the specimens was determined before and after water immersion.

### 2.5. Tensile Testing

The tensile behavior of conditioned and unconditioned BFRP and GFRP samples were tested under tensile loading following ASTM D3039. The composite samples with 200 mm length, 25 mm width and 2 mm thickness were prepared. The test was conducted using 3382 Universal Testing Machine 100 kN load cell (Instron, Norwood, MA, USA 2008) with the crosshead speed of 2 mm/min. Tensile properties data were acquired from a universal testing machine and the setup is shown in Figure 3. At least five specimens were tested for each system.

### 2.6. Flexural Testing

The flexural test was conducted according to ASTM D790 under the three-point bending test configuration as shown in Figure 4. A rectangular sample with a size of 80 mm length, 13 mm width and 3 mm thickness was prepared and the span length to thickness ratio was maintained at 16:1. A 100 kN Universal Testing Machine with the crosshead speed of 2 mm/min was used to perform the flexural test on the exposed and unexposed BFRP and GFRP samples. The flexural strength was calculated using the following equation:(1)$$\sigma_{f}=\frac{3\mathrm{PL}}{2{\mathrm{bt}}^{2}}$$
where; P: maximum load, L: support span (48 mm), b: sample width, t: sample thickness

## 3. Results and Discussion

Short and long-term exposure to environmental factors such as moisture and humidity, changing temperature, biological attack can retard the physical and mechanical properties of a composite. Moisture and temperature are the most important factors investigated, and the combination of both factors results in significant and adverse effect on the composite properties. In order to simulate the natural weathering conditions which, require assessment over a long period, test methods have been developed at an accelerated rate so that long-term weathering effects can be estimated in a shorter time.

### 3.1. Visual Observation

The visual observation of the tested specimens is captured in Figure 5, showing the surface of unmodified BFRP composites for conditioned and unconditioned specimens under the optical microscope. It can be seen that the composites experienced severe discoloration from blackish to a greyish color, due to the fiber and epoxy degradation after long exposure to water spraying and the UV radiation continuous cycle. Effective coatings or paints are suggested to be applied on the composites to prevent the fading effect and enhance the usefulness for exterior application [20]. Figure 5b shows that the weathered specimens experienced erosion at the edges and void formations because of the degradation mechanism due to UV and moisture response. The repetitive process of water spraying leached away a thin surface of the chemically modified epoxy layer on the composite and left the surface with a fresh layer which was then attacked by UV exposure, and the cycles are repeated, yielding a significant erosion of the BFRP composite surface. Other than that, the existence of water molecules (consists of OH and H ions) also triggered the photo-oxidation response which contributed to the fading effect. The main damage mechanism for a polymer subjected to UV radiation, called the photo-degradation mechanism, was initiated by UV photon absorption by catalyst residues, hydroperoxides, carbonyls and unsaturated molecules in the polymer. The UV photons excited states trigger the activation process in macromolecules leading to the physical and mechanical properties such as surface discoloration, loss of surface gloss, yellowing, as well as the formation of thin layer called chalking that leads to microcracking, fiber splitting and resin flaking in the composite, as illustrated in Figure 5b.

### 3.2. Moisture Absorption Behavior

The hydrophilic nature of cellulose-based natural fiber could limit their applications due to high moisture absorption. However, since the BF is a mineral-based natural fiber, the water absorption behavior of this type of fiber was investigated to identify its behavior towards the water. Table 2 shows the moisture absorption and swelling percentage of unconditioned and conditioned BFRP epoxy composite specimens. It can be deduced that the water absorption of the conditioned specimens was higher than the unconditioned specimens. This was due to the continuous moisture absorption during the water spraying in the accelerated weather chamber together with the water immersion process indicating the significant effect of water towards the BF composite despite its mineral-based content. Other than contributing to the photo-oxidation, the water molecules also penetrated the resin and fiber which then increased the final weight of the specimen. 

It was also observed that for both systems, the moisture absorption behavior of BF modified with nanofillers (BF-G0.1 and BF-N5) was higher than the unmodified composite. The water absorption of BF modified with GNP was 17% less than the unmodified BF, meanwhile, the BF modified with nanosilica is 11% less, for the unconditioned system. For the conditioned system, the BF modified with GNP is 14% less than the unmodified BF, meanwhile, the BF modified with nanosilica is 4% less than the unmodified BF. The reason behind this is the hydrophobic nature of the GNP and nanosilica used in the epoxy, which reduced the water molecule penetration and water absorption in the composite specimens. By comparing the moisture absorption percentage of BFRP to the GFRP composites, it can be deduced that both composite types exhibit slightly similar moisture content absorption behavior for both conditioned and unconditioned specimens. These two types of composites were prepared using the same type of resin or matrix system, therefore the behavior of moisture uptake was almost the same. This may have been due to the presence of a micro void on the specimen’s surface and between the fiber and polymer matrix that contributed to the formation of channel for moisture or water uptake in the composite. 

### 3.3. Tensile Properties

The comparison in tensile properties between the unconditioned and conditioned composites are plotted in Figure 6. From the figure, it can be observed that the loss in tensile modulus is small for the FRP composites. After exposure for 504 h to the accelerated weather conditions, the tensile modulus was reduced by 2% for unmodified BFRP and 9% for unmodified GFRP composite. The modulus of nanosilica-modified FRP composite was reduced by 0.6% for BF-N5 and 1% for GF-N5. Meanwhile, the modulus of FRP with GNP showed a reduction of 0.4% and 0.5% for BF-G0.1 and GF-G0.1 composites, respectively. The reduction in tensile modulus for the nanomodified system showed lower values than the loss for unmodified specimens, indicating that the existence of nanofiller could reduce the deterioration due to the weathering effect of the composites. A similar trend was also observed in tensile strength and strain at break, except for the tensile strain at break values for FRP-G0.1 which showed some enhancement in tensile strain. The reason behind this needs to be further investigated by conducting the accelerated weather condition test on other GNP loading. For tensile strain at break values of the accelerated weather condition specimens, it could be seen that the embrittlement of the conditioned specimens due to UV radiation resulted in a reduction in tensile strain at break. This is due to the photo-oxidation response where the diffusion of O_2_ in the specimen’s surface leads to microcrack formation which then propagates into the specimen and increases the surface degradation of the composites. Hence, when the specimen was tested under tensile loading, the failure originated from these microcracks resulting in larger deformation or decrement in tensile strain at break value. However, it can be seen that for BF-G0.1 and GF-G0.1 systems exposed to the accelerated conditions, the tensile strain at break values increased. This is attributed to the affinity to moisture behavior of graphene itself, where hydrophobicity of graphene depends on the thickness of the graphene layer used [34]. In this study, the GNP used was the multiple layer graphene which contributed to the hydrophobicity of the GNP, and this is proven by the data obtained in Table 2 where the moisture absorption and swelling percentage of the GNP-modified system showed the lowest values for both conditioned and unconditioned. Hence, due to the mechanism occurring during the elongation process, the GNP-modified system was able to resist the deformation and absorb much more strain energy before it broke, resulting in higher tensile strain at break values. 

In order to analyze the condition of the specimens’ fractured surfaces, microscopic and macroscopic observation was carried out on the postfracture specimens. This was important as the proof to support the results that have been discussed in the previous section. The interfacial failure started to occur during the nonlinear deformation, where the matrix cracked and propagated along the fiber length for both BFRP and GFRP. The macroscopic observation on the BFRP and GFRP composites are shown in Figure 7. From observation, it can be seen that both types of composite material experienced a similar damage mechanism where the matrix cracked along the fiber direction; for the conditioned specimens, the fiber breakages were much more notable than the unconditioned specimens. This is supported by the macroscopic observations by Scanning Electron Microscope (SEM). From the SEM micrographs, it could be seen that the smooth surfaces of the broken glass fibers proved that it failed through brittleness. Meanwhile, the broken surface of the basalt fiber showed that this fiber failed through a combination of brittleness and ductile behavior, where the crack on the fiber tended to elongate before completely breaking. This is the reason behind the properties of basalt fiber whereby it tends to fail at a higher tensile strain and can withstand higher tensile stress until completely fractured as compared to glass fiber.

The degradation in tensile properties of the composite was due to the fiber degradation and reduction in fiber/matrix interfacial bonding. The continuous water spraying and UV radiation cycles in the chamber affected the constituents in the fiber. The decrement in tensile properties for BFRP composites was smaller than GFRP, indicating that this type of fiber has better UV and moisture resistance. This may be due to the existence of minerals such plagioclase in basalt that help this material to sustain in extreme conditions [23]. This finding is parallel to a study by L. Yan et al. [35] where accelerated weather conditions reduced the tensile and flexural properties of flax/epoxy composites. Figure 8 summarized the effect of accelerated weather on the tensile properties of BFRP and GFRP composites. 

### 3.4. Flexural Properties

The flexural properties of unconditioned and conditioned BFRP and GFRP composites are represented in Table 3. For flexural properties, the unconditioned GFRP composite shows a lower strength value compared to BFRP by 31 MPa. All the weathered composite specimens experienced strength and modulus reduction due to UV exposure which degraded the properties of the materials. The flexural strength reduction after exposure to UV and water spraying cycles was 7% for unmodified BFRP and 8.2% for unmodified GFRP composites. The flexural strength of weathered BF-G0.1 and GF-G0.1 experienced reduction by 3.6% and 4.3%, meanwhile BF-N5 and GF-N5 reduced by 3.7% and 4.2%, respectively. This implies that the weathering had an adverse degradation effect on the flexural strength, especially for unmodified FRP, where the addition of nanofiller into the epoxy not only strengthened the composites but also provided extra protection to the specimens’ surface against the degradation. Figure 9 depicted the fractured specimens under the three-point bending test. It can be seen that similar damage mechanisms occurred for both unconditioned and conditioned specimens; matrix cracking and fiber breakage. However, the damage was much more obvious in the conditioned specimens due to the matrix degradation on the specimen surface.

A similar pattern was also observed for the flexural modulus where the unmodified BFRP and GFRP composites experienced greater decrement as compared to nanomodified composites. The flexural modulus of unmodified BFRP and GFRP reduced by 2.6% and 7.5%, respectively. The smaller reduction was observed for BF-G0.1 and GF-G0.1 with 1.1% and 4.6%, while BF-N5 and GF-N5 with 2.2% and 6.7%, respectively. Moreover, in the three-point bending test, the middle part on the top surface of the specimen is the first layer that experienced the external load along the width of the samples. Hence, the deteriorated surface of the weathered specimen is one of the major factors that contributed to the failure of the system. In order to compare the flexural properties between the BFRP and GFRP composites, it could be deduced that the reduction in flexural properties for BFRP was smaller as compared to GFRP composites, as it also represented that the durability of BFRP against the weather was higher than GFRP composites. It is expected that the reduction in flexural properties is due to UV penetration and continuous moisture absorption. The microcracks, voids and abrasion of the weathered surfaces restrict the stress mobility from epoxy to fibers, resulting in lower flexural properties. It was also noted that the degradation in tensile strength was more obvious than in flexural strength. This is because of the direction of the applied load during the related test, where the fiber pull-out failure mechanism is more pronounced in tensile due to the parallel stress applied to the fiber direction [34]. 

### 3.5. Scanning Electron Microscopic (SEM) Analysis 

In order to determine the effect of accelerated weather on the mechanical properties of FRP composite systems, SEM analysis was used to observe the fracture mechanism for unconditioned and conditioned specimens. Figure 10 depicts the SEM micrographs of the tensile specimen for unweathered and weathered conditions of unmodified BFRP composites. It clearly can be seen that the unconditioned composite had a smooth surface as it was not deteriorated by UV exposure. On the other hand, the uneven rough surface could be seen on the conditioned specimen as well as the delamination and crack of the matrix which suggested weak interfacial adhesion between the matrix and fiber due to the degradation of the epoxy matrix. Figure 11 shows the SEM micrographs of unconditioned and conditioned nanosilica-modified BFRP and GFRP composite surfaces. Similar to the unmodified system, by exposing the samples to the accelerated weathering conditions, the surface became rougher and the formation of voids and cracks was much more obvious. It was found that the UV exposure on the surface caused the degradation which initiated the fractures due to the existence of stress concentrators such as voids, microcracks, fiber and matrix cracking as well as fiber fragmentation. As cracks became bigger, water and light penetration became much more feasible and facilitated the degradation of the composites.

## 4. Conclusions

In this study, the durability studies were conducted on a series of BFRP and GFRP composites under accelerated weather. The tensile, flexural and water absorption properties of the composites were evaluated for unweathered and weathered environment. SEM analysis was also performed to observe the composite microstructure and their fracture behavior. The performance of natural BFRP composites was compared to the synthetic GFRP composites. Several conclusions are drawn from this study and listed as follows:Composites underwent severe discoloration due to the weathering effect resulting from fibre and matrix degradation after exposure to water spraying and UV radiation cycles.The moisture absorption of FRP composites modified with nanofillers was lower than for the unmodified composites because of the hydrophobic nature of the GNP and nanosilica used which reduced water molecule penetration and water absorption into the composite specimens.Significant reduction was observed in tensile and flexural properties of unmodified BFRP composites where the tensile strength and modulus for the weathered specimen was reduced by 12% and 2%, respectively. The reduction in flexural strength and modulus was 7% and 2.6%, respectively, indicating the effects of weathering conditions are more pronounced in tensile than flexural properties.The reduction in tensile and flexural properties of weathered nanomodified BFRP composites was small: the tensile strength and modulus of BF-G0.1 was reduced by 0.5% for both properties, while the flexural strength and modulus were reduced by 3.6% and 12%, respectively. Meanwhile, for BF-N5, the tensile strength and modulus were reduced by 7% and 0.7%, respectively, while flexural strength and modulus was reduced by 3.7% and 4.7%, respectively.The reduction in tensile and flexural properties of the nanomodified system showed lower values than the loss for unmodified specimens, indicating that the existence of nanofiller could reduce deterioration due to weathering of the composites.The UV exposure on the composites decreased the composites’ mechanical properties due to surface deterioration as revealed by SEM analysis.The reduction in tensile and flexural properties for BFRP composites was smaller than for GFRP, indicating that this type of fibre had better UV and moisture resistance due to the existence of a mineral such plagioclase in basalt that helps this material to withstand extreme conditions.

Based on the findings of this study, the natural BFRP composite has the potential to be used in related applications such as structural material, due to its comparable properties to the synthetic GFRP composite. However, it is advised to apply a proper treatment or coating to the fiber to enhance its durability performance for wide use in many applications, especially in the construction industry.

## Figures and Tables

**Figure 1 polymers-12-02621-f001:**
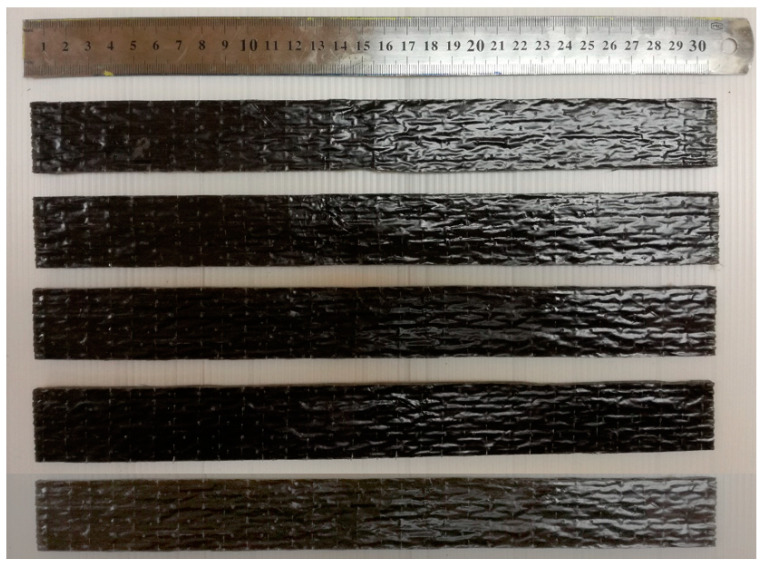
Examples of fabricated unmodified basalt fiber reinforced polymer (BFRP) composite samples.

**Figure 2 polymers-12-02621-f002:**
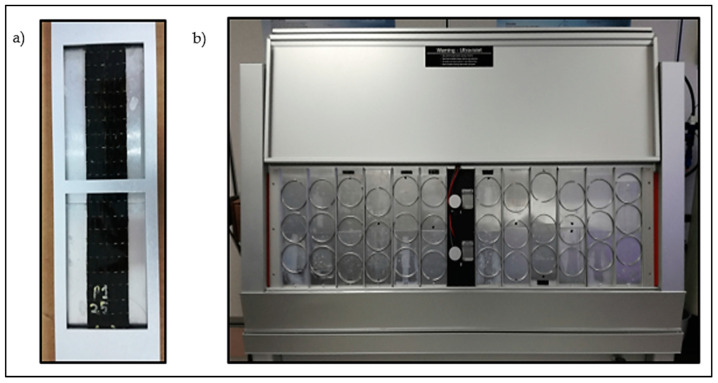
(**a**) Specimen mounted on the specimen holder, (**b**) setup of the specimen holder arrangement in the accelerated weather testing chamber.

**Figure 3 polymers-12-02621-f003:**
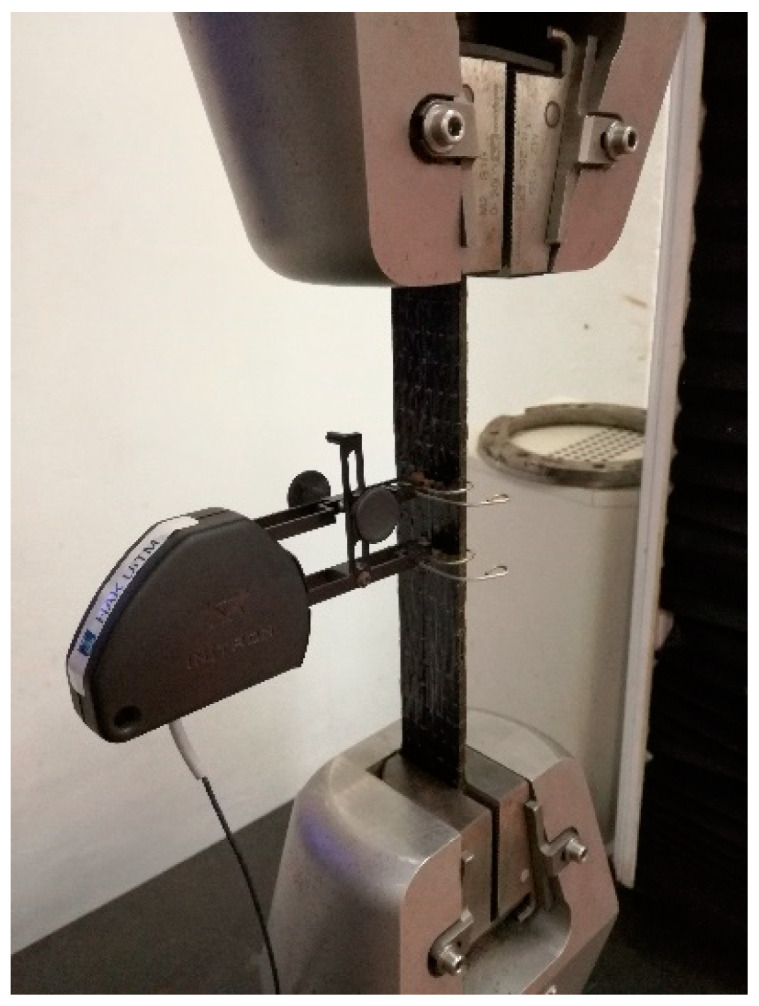
Tensile test setup.

**Figure 4 polymers-12-02621-f004:**
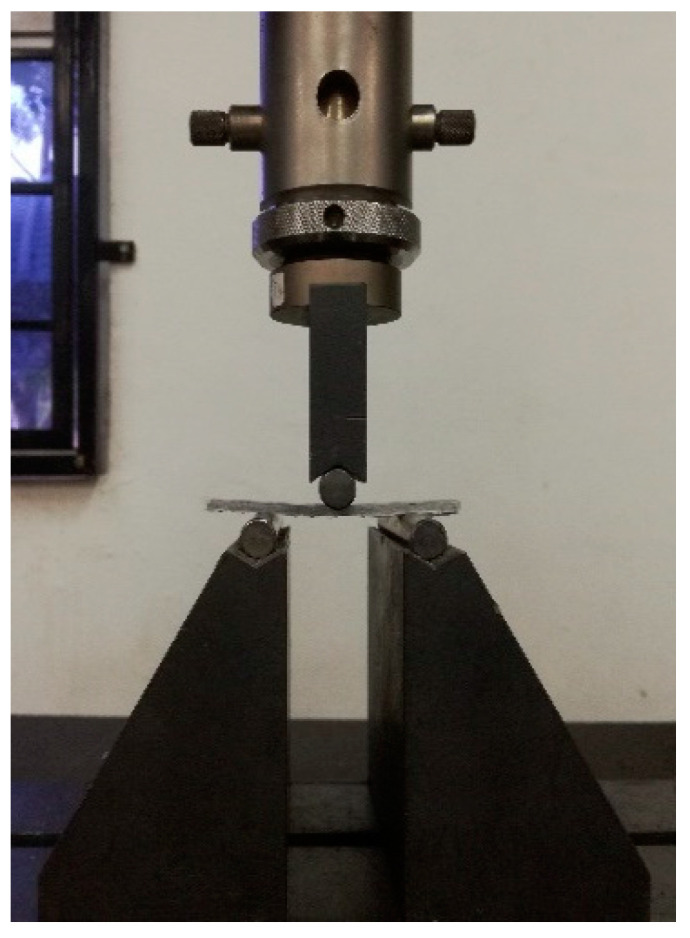
Three-point bending test on the fiber reinforced polymer (FRP) composites.

**Figure 5 polymers-12-02621-f005:**
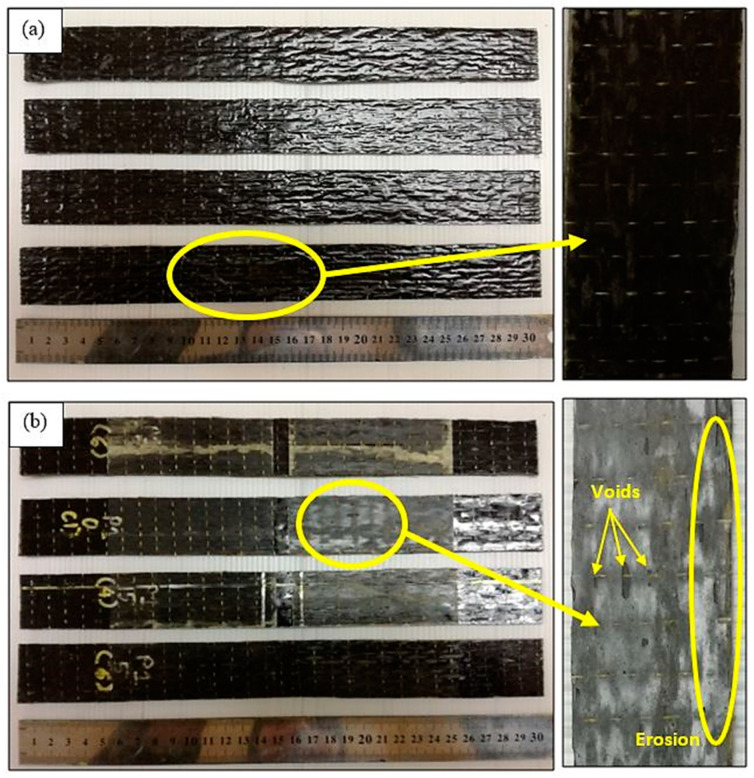
Overall surface of unmodified BFRP composites (**a**) unconditioned, (**b**) conditioned specimens.

**Figure 6 polymers-12-02621-f006:**
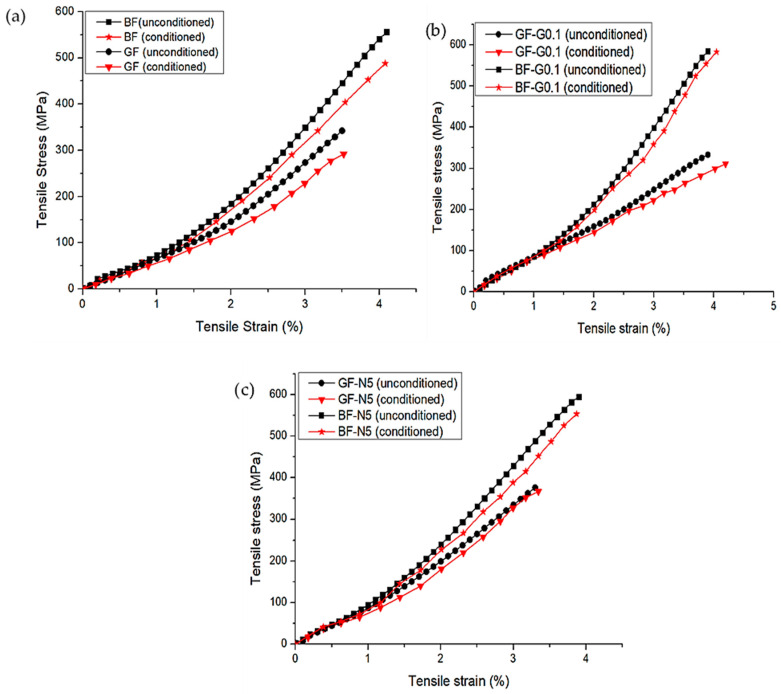
Tensile stress−strain response of unmodified and nanomodified specimens for conditioned and unconditioned systems of (**a**) unmodified FRP composites, (**b**) GNP modified FRP composites, and (**c**) Nanosilica modified FRP composites.

**Figure 7 polymers-12-02621-f007:**
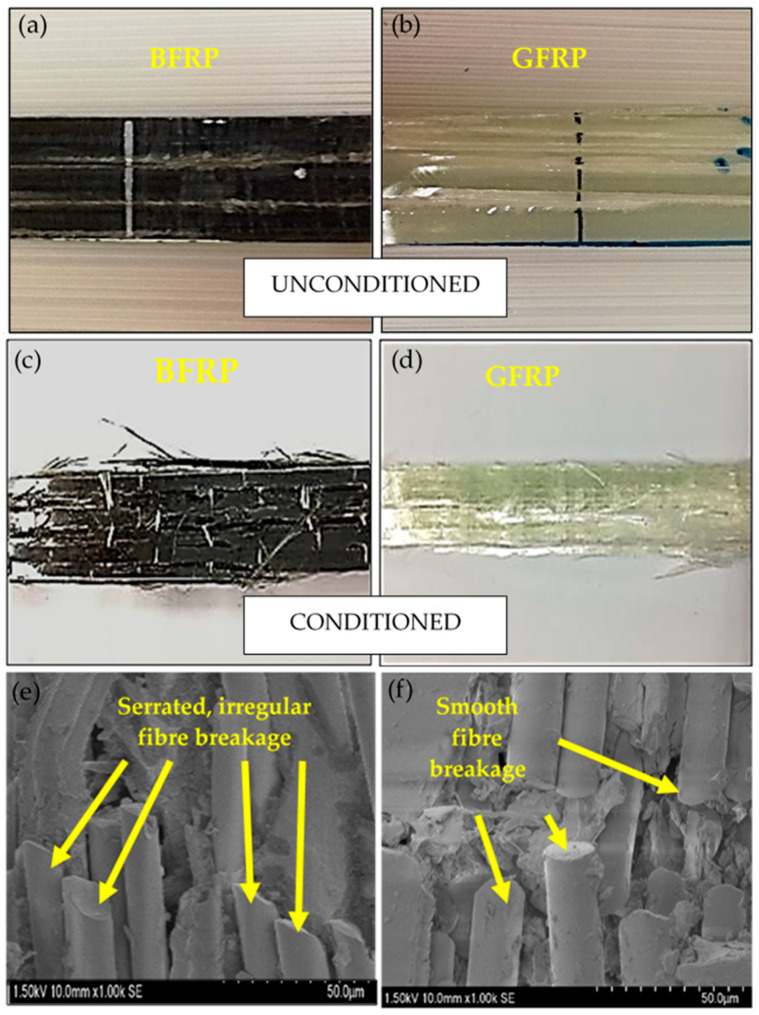
Fractured specimens of unmodified composites under tensile loading for (**a**,**b**) unconditioned, (**c**,**d**) conditioned and, (**e**,**f**) SEM micrographs of the damaged specimens.

**Figure 8 polymers-12-02621-f008:**
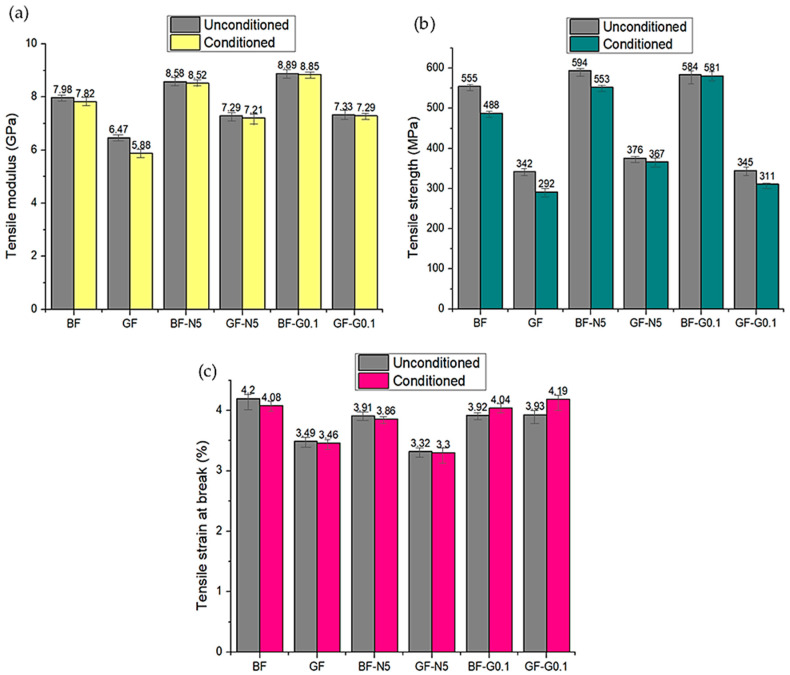
Effect of accelerated weather conditions on the (**a**) tensile modulus, (**b**) tensile strength, and (**c**) tensile strain at break of BFRP and GFRP composites.

**Figure 9 polymers-12-02621-f009:**
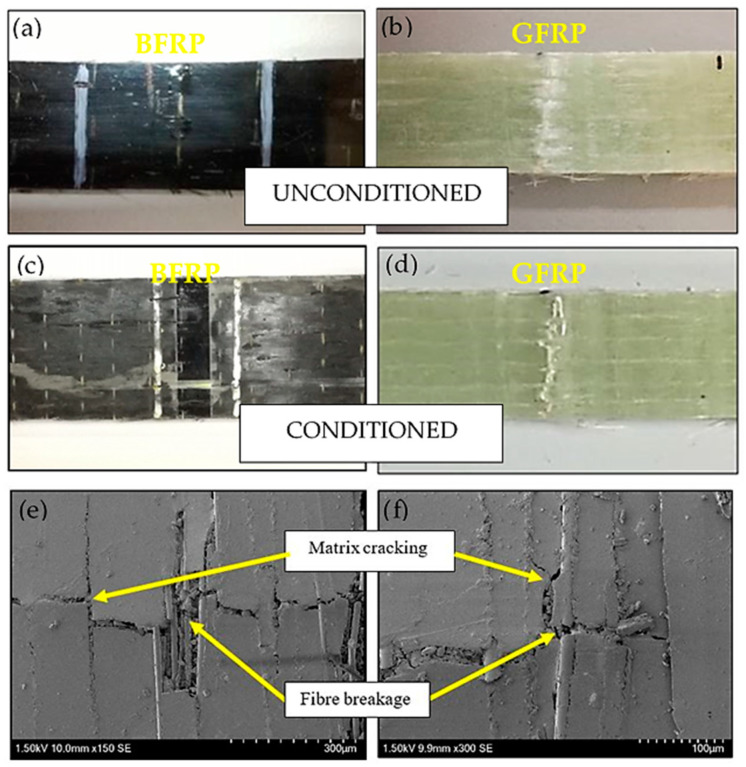
Fractured specimens of unmodified composite under three-point bending, where (**a**,**b**) unconditioned, (**c**,**d**) conditioned and, (**e**,**f**) SEM micrographs of the damaged specimens.

**Figure 10 polymers-12-02621-f010:**
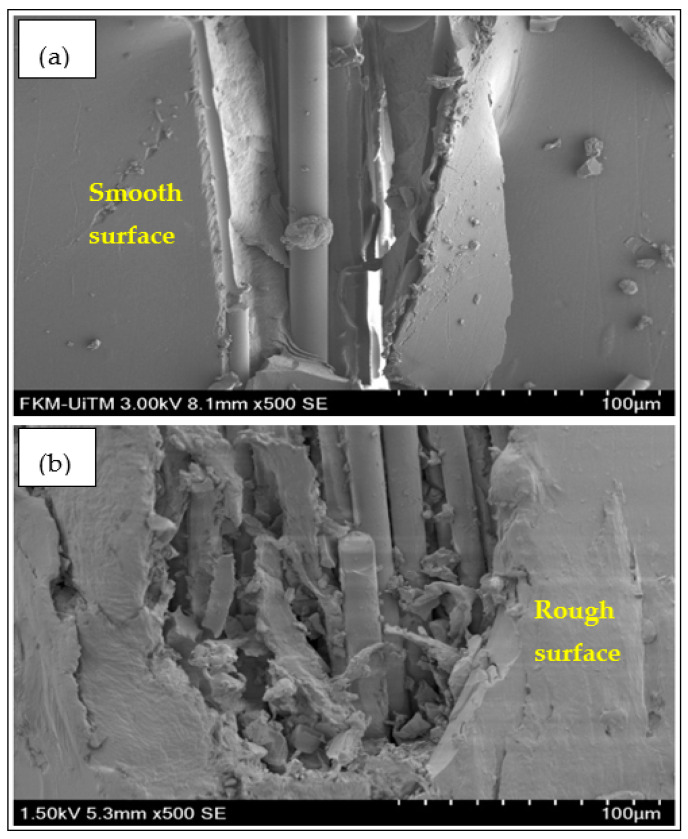
SEM micrographs of fractured unmodified BFRP composites of (**a**) unconditioned and (**b**) conditioned specimen.

**Figure 11 polymers-12-02621-f011:**
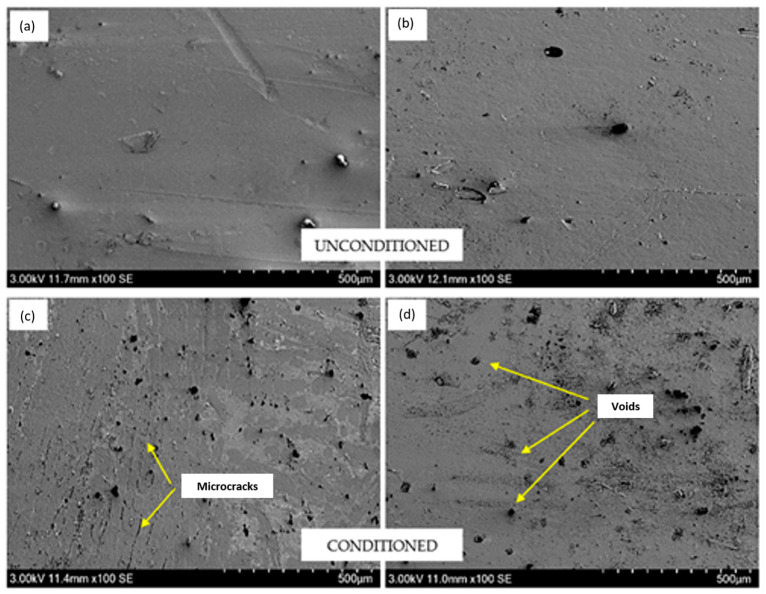
SEM micrographs of unconditioned nanosilica modified (**a**) BFRP and (**b**) GFRP and conditioned nanosilica modified (**c**) BFRP and (**d**) GFRP composites surfaces.

**Table 1 polymers-12-02621-t001:** Specimen labels and details.

Specimen Label	Specimen Details
BF	Unmodified basalt fiber
GF	Unmodified glass fiber
BF-G0.1	BF with 0.1 wt.% GNP
GF-G0.1	GF with 0.1 wt.% GNP
BF-N5	BF with 5 wt.% nanosilica
GF-N5	GF with 5 wt.% nanosilica

**Table 2 polymers-12-02621-t002:** Water absorption and swelling behavior of BFRP composites.

System	Samples	Moisture Absorption Percentage after 24 h (%)	Swelling Percentage after 24 h (%)
Unconditioned	BF	0.34 ± 0.15	0.05 ± 0.02
	BF-G0.1	0.28 ± 0.23	0.03 ± 0.03
	BF-N5	0.30 ± 0.11	0.03 ± 0.06
	GF	0.28 ± 0.20	0.03 ± 0.06
	GF-G0.1	0.24 ± 0.24	0.02 ± 0.08
	GF-N5	0.30 ± 0.21	0.04 ± 0.05
Conditioned	BF	1.32 ± 0.21	0.32 ± 0.07
	BF-G0.1	1.13 ± 0.15	0.17 ± 0.08
	BF-N5	1.26 ± 0.12	0.28 ± 0.10
	GF	1.34 ± 0.18	0.17 ± 0.20
	GF-G0.1	1.18 ± 0.22	0.12 ± 0.11
	GF-N5	1.22 ± 0.27	0.29 ± 0.16

**Table 3 polymers-12-02621-t003:** The variation in flexural properties of BFRP and GFRP composites under accelerated weather conditions.

System	Sample	Flexural Strength (MPa)	Flexural Modulus (GPa)
Unconditioned	BF	417.57 ± 5.3	9.18 ± 2.8
	BF-G0.1	432.45 ± 4.2	10.56 ± 3.1
	BF-N5	428.23 ± 3.9	9.84 ± 3.6
	GF	386.26 ± 9.1	8.24 ± 3.7
	GF-G0.1	395.22 ± 6.7	9.04 ± 1.8
	GF-N5	391.89 ± 5.4	8.77 ± 2.9
Conditioned	BF	388.24 ± 7.2	8.94 ± 4.2
	BF-G0.1	416.78 ± 5.6	9.21 ± 3.5
	BF-N5	412.51 ± 3.8	9.38 ± 2.8
	GF	354.32 ± 7.7	7.62 ± 5.3
	GF-G0.1	377.98 ± 9.1	8.62 ± 1.9
	GF-N5	375.21 ± 2.9	8.19 ± 2.6

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
