# Peer review of "Effects of Accelerated Weathering on Degradation Behavior of Basalt Fiber Reinforced Polymer Nanocomposites"

_polymers, 2020, doi:10.3390/polym12112621_

Round 1
Reviewer 1 Report
The paper presents an interesting work on the basalt fibre reinforced polymer composites and the effect of accelerated weathering condition (UV irradiation and water spraying) on their mechanical and physical properties. The results and analyses are both interesting and valuable. As such, the paper may be of interest to the readers of Polymers. However, the article has several deficiencies and it needs some corrections before being considered for publication:
(1) The chemical contents of basalt fibres may be different because of their producing areas. Is the effect of the basalt fibres on the mechanical and physical properties of composites (under and not under accelerated weather condition) significant?
(2) Why did the tensile strain at break of BF-G0.1 and GF-G0.1 increase under accelerated weather condition?
(3) Using both GNP and nanosilica to modify the fibres can enhance the mechanical and physical properties of composites effectively. How did the author determine the dosages of GNP and nanosilica? Why not increase the dosages of them?
(4) The modification enhanced the composites’ properties under and not under accelerated weather condition, but it is not clear that the modification affected the mechanical properties or the durability of composites. The author should discuss the mechanism of the durability improvement.
Author Response
Are the results clearly presented? Can be improved
We improved as per your suggestion
Are the conclusions supported by the results? Can be improved
We improved as per your suggestion
Comments and Suggestions for Authors
The paper presents an interesting work on the basalt fibre reinforced polymer composites and the effect of accelerated weathering condition (UV irradiation and water spraying) on their mechanical and physical properties. The results and analyses are both interesting and valuable. As such, the paper may be of interest to the readers of Polymers. However, the article has several deficiencies and it needs some corrections before being considered for publication:
(1) The chemical contents of basalt fibres may be different because of their producing areas. Is the effect of the basalt fibres on the mechanical and physical properties of composites (under and not under accelerated weather condition) significant?
Thank you for the comments.
The effects of basalt fibre on the mechanical and physical properties of composites for both conditioned and unconditioned composites are highly significant to be studied and reported. Based on the study, the mechanical properties of Basalt composites are better than those of conventional glass fibre composites in both unconditioned and conditioned samples. Further improvement on the properties of basalt composites was observed when nanographene and nanosilica were added into the composite system.
In this paper, the effects of weathering condition on tensile and flexural properties of BFRP and conventional GFRP composites were studied and reported. In general, the results showed the UV exposure and water absorption caused reduction in tensile and flexural strength of the composites.
(2) Why did the tensile strain at break of BF-G0.1 and GF-G0.1 increase under accelerated weather condition?
The tensile strain at break for BF-G0.1 and GF-G0.1 increased under accelerated weather condition attributed to the affinity to moisture behaviour of graphene itself, where hydrophobicity of graphene depends on the thickness of the graphene layer used. In this study, the GNP used was the multiple layer graphene which contributed to the hydrophobicity of the GNP, and this is proven by the data obtained in Table 2 where the moisture absorption and swelling percentage of GNP modified system showed the lowest values for both conditioned and unconditioned. Hence, due to the mechanism occurred during the elongation process, the GNP modified system able to resist the deformation and absorb much more strain energy before it breaks resulting in higher tensile strain at break values.
(3) Using both GNP and nanosilica to modify the fibres can enhance the mechanical and physical properties of composites effectively. How did the author determine the dosages of GNP and nanosilica? Why not increase the dosages of them?
A preliminary study was conducted by the authors before to select the optimum filler loading that works best in the epoxy matrix based on the experimental work and also refers to the literature survey. An optimum nanofiller loadings of 0.1wt% GNP and 5wt% nanosilica were selected and used in this study, in order to ensure homogeneous dispersion of nanofiller in FRP.
(4) The modification enhanced the composites’ properties under and not under accelerated weather condition, but it is not clear that the modification affected the mechanical properties or the durability of composites. The author should discuss the mechanism of the durability improvement.
The paper has been revised in which the discussion of the degradation mechanisms, mechanical and durability of composites has been improved as suggested by the reviewer. (page 6, line 21), (page 8, line 250), (page 8, line 273).
Reviewer 2 Report
This work aims to give insight on the effect of accelerated weathering, i.e. combination of Ultra-Violet (UV) exposure and water spraying, on the visual and mechanical properties of Basalt Fibre Reinforced Polymer (BFRP) composites, in which unmodified BFRP, nanosilica modified BFRP and Graphene Nanoplatelet (GNP) modified BFRP composites laminates were fabricated, Glass Fibre Reinforced Polymer (GFRP) laminate was also prepared for performance comparison purposes between the natural and synthetic fibres. It is a novel paper and worthy to investigate deeply. However, the assessments on the mechanical response of BFRP and GFRP composite laminates with different constitute and under different conditions are not elaborated clearly and discussed in detail. The conclusions need to be enhanced as well. Some advices are shown as follows:
- In Page 3 line 142, “A total of five samples per group 142 of FRP composites were fabricated for the testing purpose”. It is better to show the photos of these samples.
- In Page 5, 2.5 Tensile testing and 2.6 Flexural testing. It is better to exhibit the photos of samples under tensile and flexural testing.
- In Page 7, line 248. Since it is a whole comparison between two systems of unmodified and nanomodified composite specimens with condition as well as un-condition. Where is the table showing the moisture absorption and swelling percentage of unconditioned and conditioned GFRP epoxy composites specimens?
- In Page 7, there is a mistake in line 272.
- In Page 9, no standard deviation bars are shown in Fig. 4, since they are repeated experimental tests.
- For 3.3 Tensile properties and 3.4 Flexural properties. Some photos of fractured specimens should be added and elaborated more to enhance the depth of the discussion.
- In Page 11, line 343, there is a mistake of Fig. 4.
- The discussion in 3.5 Scanning Electron Microscopic (SEM) analysis should be illustrated in detail. Additionally, where are the SEM micrographs of fractured modified BFRP (or GFRP) composites of unconditioned and conditioned specimen?
- The conclusions should be enhanced and discussed deeply. It is confused that where is the conclusion about the tensile and flexural properties of modified BFRP composites (BF-G0.1 and BF-N5)?
Author Response
English language and style are fine/minor spell check required
We checked and rectify where needed.
Are the methods adequately described? Can be improved
Yes We made changes.
Are the results clearly presented? Can be improved
Yes We made changes.
Are the conclusions supported by the results? Can be improved
Yes We made changes.
Comments and Suggestions for Authors
This work aims to give insight on the effect of accelerated weathering, i.e. combination of Ultra-Violet (UV) exposure and water spraying, on the visual and mechanical properties of Basalt Fibre Reinforced Polymer (BFRP) composites, in which unmodified BFRP, nanosilica modified BFRP and Graphene Nanoplatelet (GNP) modified BFRP composites laminates were fabricated, Glass Fibre Reinforced Polymer (GFRP) laminate was also prepared for performance comparison purposes between the natural and synthetic fibres. It is a novel paper and worthy to investigate deeply. However, the assessments on the mechanical response of BFRP and GFRP composite laminates with different constitute and under different conditions are not elaborated clearly and discussed in detail. The conclusions need to be enhanced as well. Some advices are shown as follows:
- In Page 3 line 142, “A total of five samples per group 142 of FRP composites were fabricated for the testing purpose”. It is better to show the photos of these samples.
Thank you for the comments and recommendations given by the reviewer. Photos of fabricated samples have been added as suggested by the reviewer as in Figure 1 (page 4).
2. In Page 5, 2.5 Tensile testing and 2.6 Flexural testing. It is better to exhibit the photos of samples under tensile and flexural testing.
Photos of samples under tensile and flexural tests have been added as suggested in Figure 3 (page 5) and Figure 4 (page 6).
3. In Page 7, line 248. Since it is a whole comparison between two systems of unmodified and nanomodified composite specimens with condition as well as un-condition. Where is the table showing the moisture absorption and swelling percentage of unconditioned and conditioned GFRP epoxy composites specimens?
The table has been changed to a new one by inserting the data for GFRP composites for both unconditioned and conditioned specimens together with the related discussion.
4. In Page 7, there is a mistake in line 272.
The mistake in line 272 has been corrected accordingly. (line 297, page 9).
5. In Page 9, no standard deviation bars are shown in Fig. 4, since they are repeated experimental tests.
Figure 4, now Figure 8 has been replotted with standard deviation bars.
6. For 3.3 Tensile properties and 3.4 Flexural properties. Some photos of fractured specimens should be added and elaborated more to enhance the depth of the discussion.
Photos of fractured specimens have been added with the details to enhance the depth of the discussion (Figure 7, page 10), (Figure 9, page12).
7. In Page 11, line 343, there is a mistake of Fig. 4.
Figure 4 has been changed to Figure 10 and the explanation in text has been updated accordingly (page 13, line 374).
8. The discussion in 3.5 Scanning Electron Microscopic (SEM) analysis should be illustrated in detail. Additionally, where are the SEM micrographs of fractured modified BFRP (or GFRP) composites of unconditioned and conditioned specimen?
The SEM micrographs of fractured nanomodified BFRP and GFRP composites of unconditioned and conditioned specimen have been added as in Figure 11 (page 14) with details.
9. The conclusions should be enhanced and discussed deeply. It is confused that where is the conclusion about the tensile and flexural properties of modified BFRP composites (BF-G0.1 and BF-N5)?
The conclusion has been improved by addition of the tensile and flexural properties of nanomodified BFRP composites.
Reviewer 3 Report
I accept the paper in its present form.
Author Response
English language and style are fine/minor spell check required
We improved
Comments and Suggestions for Authors
I accept the paper in its present form.
Thanks for your acceptance.